# Silver Catalyzed Decarbonylative [3 + 2] Cycloaddition of Cyclobutenediones and Formamides

**DOI:** 10.3390/molecules26102974

**Published:** 2021-05-17

**Authors:** Pengcheng Wang, Ruirui Yu, Sajjad Ali, Zhengshen Wang, Zhigang Liu, Jinming Gao, Huaiji Zheng

**Affiliations:** Shaanxi Key Laboratory of Natural Products and Chemical Biology, College of Chemistry and Pharmacy, Northwest Agriculture and Forestry University, 3 Taicheng Road, Yangling 712100, China; 18392425641@163.com (P.W.); yrr341602@nwafu.edu.cn (R.Y.); sajjadali_awan@yahoo.com (S.A.); lzg2013@nwsuaf.edu.cn (Z.L.); jinminggao@nwsuaf.edu.cn (J.G.)

**Keywords:** *N*,*O*-acetal, cyclobutenediones, formamides, [3 + 2] cycloaddition

## Abstract

As an important moiety in natural products, *N*,*O*-acetal has attracted wide attention in the past few years. An efficient method to construct *N*,*O*-acetal has been developed. Using silver catalyst, cyclobutenediones were smoothly converted to the corresponding *γ*-aminobutenolides in the presence of formamides, in which cyclobutenediones likely proceed with a key decarbonylative [3 + 2] cycloaddition process. In this way, a series of products with varied substituents were isolated in moderate yield and fully characterized.

## 1. Introduction

*N*,*O*-Acetal/ketal represent an important architecture in bioactive natural and pharmaceutical products [1,2,3]. Typical structures such as aspidosperma alkaloid (-)-aspidophytine [4,5], lycopodium alkaloid lycoplanine A [6], indoline alkaloid rosibiline [7], and macrocyclic marine alkaloid ‘upenamide [8,9] are shown in Figure 1. Besides, Tegafur [10] containing a hemiaminal ether skeleton is used as an anticancer reagent and indolinooxzolidines [11] are effective in molecular switching.

Many efforts [12,13] have been made to achieve the *N*,*O*-acetal/ketal moiety including: (1) acetalization/ketalization of oxonium or imine ion (eq. 1, Scheme 1) [14,15], (2) C-H bond activation of ether with various nitrogen reagents (eq. 2) [16,17,18,19,20,21], and (3) cycloaddition of dipole with or derived from amide (eq. 3) [22,23]. In this respect, Sun group reported a type of [3 + 2] cycloaddition to achieve 5-aminofuran-2(5*H*)-one using cyclopropenone and amide under Ag catalyst [24]. Similar work was reported by Matsuda and co-workers [25].

As part of the transformation of small ring compounds, especially squaric acid in our group, we wish to establish an alternative approach for the construction of *N*,*O*-acetal from squaric acid or other four-membered cyclic compounds. Although the transformation of squaric acid and other cyclobutenediones to cyclopropenones under photolysis is known [26,27,28], the investigation of their thermal stability is still lacking. Based on this, we anticipated that the strained ring compound cyclobutenediones [29,30,31] could proceed with a decarbonylation process under metal-catalyst to form cyclopropenone intermediate similar to their photochemical nature, and thus achieve *N*,*O*-acetal/ketal skeleton.

## 2. Results

We initiated our studies by probing various reaction conditions for the cycloaddition of diphenylcyclobutenedione [32] (**1a**) with *N*,*N*-dimethylformamide (DMF, **2a**), and the results are given in Table 1. Heating the reaction mixture at 170 °C in the absence of catalyst, no reaction occurred (Table 1, entry 1). Considering the wide catalytic applications of transition metals in carbonylation and/or decarbonylation reaction, rhodium [33] and palladium [34] salts were firstly adopted in our decarbonylative [3 + 2] cycloaddition. Fortunately, the reaction occurred under the rhodium catalyst [Cp*RhCl_2_]_2_ or Rh(PPh_3_)_3_Cl (Entries 2 and 3). Due to the low yield of **3aa** under rhodium catalyst, we thus turn our attention to the palladium catalyst such as Pd(OAc)_2_ or Pd(PPh_3_)_4_. It was found that Pd(II) leads to the decomposition of cyclobutenedione, and Pd(0) catalyzes the formation of **3aa** (Entries 4 and 5).

Based on these results, other transition metals without obvious coordination effect with CO were screened. Most of the metals yielded similar results as Cu(OTf)_2_, and gave no desired product during the reaction (Table 1, entry 6). According to the [3 + 2] cycloaddition of cyclopropenone, various Ag salts were then investigated and proved effective for this transformation. When AgSbF_6_ was employed as the catalyst, the desired product **3aa** was isolated in only 21% yield (Entry 7). Further examination demonstrated that the reaction proceeded most efficiently with AgNTf_2_ catalyst and the product **3aa** was offered in 60% yield (Entries 8–10). It should be mentioned that longer reaction time proved ineffective for this transformation (Entry 11). Moreover, no annulation reaction took place upon lowering the temperature to 160 °C (Entry 12). Other solvents such as chlorobenzene were also investigated for the reaction and the product **3aa** was obtained in 52% yield, which was slightly lower than *o*-dichlorobenzene (Entry 13). Copies of ^1^H and ^13^C NMR spectra for **3aa** and its crystallographic data are available in the Appendix A.

With the optimized [3 + 2] cycloaddition conditions in hand, the scope and generality of the annulation reactions for the formation of products **3** were then investigated (Table 2). We firstly examined the reaction of various formamides (**2a**–**f**) with diphenylcyclobutenedione (**1a**) under standard conditions. The experiment results demonstrated that the steric hinerance of formamides had an obvious effect on the reaction. Compared with the yield of *N*,*N*-dimethylformamide (**2a**), the coupling product (**3ab**) of *N*,*N*-diethylformamide (**2b**) with **1a** was isolated in only 36% yield. In addition, the extension of *N*-methyl-*N*-phenylformamide (**2c**) was unsuccessful, and only a few products (**3ac**) could be detected. Although the increasingly steric hindrance led to the lower yield, the various cyclic formamides (**2d**–**2f**) could be converted smoothly to give the corresponding products (**3ad**–**3af**) in moderate yield.

Further exploration demonstrated that various aromatic substituted cyclobutenediones were suitable for this reaction (Table 2). As indicated, diarylcyclobutenediones bearing electron-donating groups on the phenyl rings such as 4-methyl, 4-ethyl and 4-methoxy, respectively, provided the corresponding annulation products (**3ba**, **3ca** and **3da**) in higher yields, thus broadening the application of current methodology. Meanwhile, this transformation can be extended to the halogen substituted cyclobutenediones, although the isolated yields of products (**3ea**, **3fa** and **3ga**) were decreased slightly. Moreover, when the position of the substituent on the phenyl rings changed, the corresponding products (**3ha**, **3ia**) were still isolated in good yields.

To further expand the application of the Ag-catalyzed [3 + 2]-cycloaddition reaction, cross experiments with various substituent groups on formamides or the phenyl rings were screened. Results are given in Table 3; ten new corresponding annulation products were obtained in moderate yield.

Finally, the derivatization experiments of [3 + 2]-annulation product were investigated. *N*,*O*-Acetal was easily converted to hemiacetal under acid condition. Upon heating the product **3aa** in THF at 65 °C in the presence of H_2_SO_4_, **4** was isolated in 66% yield, which could be further alkylated to **5** or **6** using Grignard reaction or HWE olefination [35], respectively (Scheme 2).

A possible mechanism for the decarbonylative [3 + 2] cycloaddition of cyclobutenedione 1a with formamide 2a is tentatively proposed. As shown in Scheme 3, following the initial chelation of Ag^+^ with the carbonyl group to generate A, the ring-reducing process is occurred and lead to the formation of intermediate B [36]. After the extrusion of CO, the cyclopropenone intermediate C is formed. The next steps involving ring-opening, nucleophilic addition of formamide, and regeneration of silver catalyst are the same with the literatures [24,25].

## 3. Materials and Methods

### 3.1. General Methods

Unless otherwise stated, all reactions were carried out under argon atmosphere. All commercial available reagents (Energy Chemical, Shanghai, China) were used without further purification. Anhydrous solvents including chlorobenzene and o-dichlorobenzene were commercially available. Tetrahydrofuran (THF) was distilled from sodium. Column chromatography was performed on silica gel (200–300 mesh). ^1^H NMR spectra (Bruker, Fällanden, Switzerland) were recorded on a 500 MHz NMR spectrometer and ^13^C NMR spectra were recorded on a 125 MHz NMR spectrometer. HRMS data were recorded on Thermo Scientific LTQ Orbitrap XL. Melting points were uncorrected.

### 3.2. Experiment Procedures

General procedure for the synthesis of cyclobutenedione. THF (150 mL) was added to a mixture of Fe_2_(CO)_9_ (5.457 g, 15.00 mmol) and *t*-BuOK (2.245 g, 20.00 mmol) at room temperature under argon. The resulting mixture was stirred for 0.5 h at room temperature and another 15 min at 65 °C. Diphenylacetylene (0.892 g, 5.00 mmol) was added and then further stirred for 12 h at 75 °C. The mixture was cooled to room temperature, and CuCl_2_·2H_2_O (12.786 g, 75.00 mmol) in acetone (50 mL) was added. After filtration, the filtrate was concentrated under reduced pressure. The residue was purified by flash chromatography (petrol ether/EtOAc = 20:1) to give products. The NMR data are consistent with previous reports in the literature (**1a** [32,37], **1b**, **1d**, **1f**–**1h** [37], **1e** [38]).

General procedure for [3 + 2] cycloaddition of cyclobutenediones with formamides. A pressure tube was charged with AgNTf_2_ (0.012 g, 0.03 mmol) and 3,4-diphenylcyclobut-3-ene-1,2-dione **1a** (0.035 g, 0.15 mmol), and then was evacuated and backfilled with argon. DMF **2a** (0.25 mL, 3.00 mmol) and *o*-dichlorobenzene (1.5 mL) were added via a syringe. The mixture was heated at 170 °C for 24 h. After cooling to room temperature, the mixture was subjected to flash chromatography (petrol ether/EtOAc = 10:1) to give products. The NMR data are consistent with previous reports in the literature (**3aa**, **3ab**, **3af**, **3ba**, **3da** [24,25], **3ad, 3da** [25], **3ae** [39], **3ea**–**3ga** [24]). Copies of ^1^H and ^13^C NMR spectra for all products and crystallographic data for **3aa** are available in the Appendix A.

Procedure for the synthesis of product **4**. To a solution of **3aa** (0.144 g, 0.51 mmol) in THF (3 mL) was added 0.5 M H_2_SO_4_ (3mL). The reaction mixture was heated at 65 °C for 36 h. After cooled to room temperature, the mixture was quenched with saturated NaHCO_3_ solution. The separated layer was extracted with ethyl acetate twice. The combined organic layers were dried over anhydrous Na_2_SO_4_, and then concentrated under reduced pressure. The crude product was purified by flash chromatography (petrol ether/EtOAc = 3:1) to give product **4**. The NMR data are consistent with previous reports in the literature. [40]

Procedure for the synthesis of product **5**. To a solution of **4** (0.032 g, 0.13 mmol) in THF (3 mL) at −78 °C under argon was added a solution of allylmagnesium bromide in diethyl ether (0.39 mL, 1.0 M, 0.39 mmol) via syringe. The reaction was stirred at room temperature for 1.5 h, and then quenched with saturated NH_4_Cl solution. The separated aqueous layer was extracted with ethyl acetate twice. The combined organic layers were dried over anhydrous Na_2_SO_4_, and then concentrated. The crude product was purified by flash chromatography (petrol ether/EtOAc = 10:1) to give product **5**.

Procedure for the synthesis of product **6**. To a solution of Triethyl phosphonoacetate (0.088 g, 0.36 mmol) in THF (2 mL) at 0 °C was added NaH (0.015 g, 0.36 mmol). After 30 min, a solution of **4** (0.030 g, 0.12 mmol) in THF (1 mL) was added dropwise. The resulting mixture was stirred for 3 h, and then quenched with saturated NH_4_Cl solution. The separated aqueous layer was extracted with ethyl acetate twice. The combined organic layers were dried over anhydrous Na_2_SO_4_, and then concentrated. The crude product was purified by flash chromatography (petrol ether/EtOAc = 5:1) to give product **6**.

### 3.3. Characterization of the Products

*3,4-Diphenylcyclobut-3-ene-1,2-dione* (**1a**). Yellow solid, mp 89.8–91.2 °C; 0.878 g, yield 75%; ^1^H NMR (500 MHz, CDCl_3_) δ: 8.10–8.04 (m, 4H), 7.64–7.51(m, 6H); ^13^C NMR (125 MHz, CDCl_3_) δ: 196.07, 187.40, 133.35, 129.26, 128.15, 128.08; HRMS (ESI) *m*/*z* calcd for C_16_H_10_O_2_Na^+^ [M+Na]^+^: 257.05730, found: 257.05728.

*3,4-Di-p-tolylcyclobut-3-ene-1,2-dione* (**1b**). Yellow solid, mp 169.4–172.3 °C; 0.630 g, yield 48%; ^1^H NMR (500 MHz, CDCl_3_) δ: 7.99 (d, *J* = 8.0 Hz, 4H), 7.35 (d, *J* = 8.0 Hz, 4H), 2.46 (s, 6H); ^13^C NMR (125 MHz, CDCl_3_) δ: 196.37, 186.54, 144.39, 129.97, 128.25, 125.64, 21.97; HRMS (ESI) *m*/*z* calcd for C_18_H_14_O_2_Na^+^ [M+Na]^+^: 285.08860, found: 285.08859.

*3,4-Bis(4-ethylphenyl)cyclobut-3-ene-1,2-dione* (**1c**). Yellow solid, mp 93.2–95.6 °C; 0.653 g, yield 45%; ^1^H NMR (500 MHz, CDCl_3_) δ: 8.03 (d, *J* = 8.0 Hz, 4H), 7.37 (d, *J* = 8.0 Hz, 4H), 2.75 (q, *J* = 7.5 Hz, 4H), 1.30 (t, *J* = 7.5 Hz, 6H); ^13^C NMR (125 MHz, CDCl_3_) δ: 196.40, 186.54, 150.55, 128.78, 128.36, 125.83, 29.20, 15.08; HRMS (ESI) *m*/*z* calcd for C_20_H_18_O_2_Na^+^ [M+Na]^+^: 313.11990, found: 313.11984.

*3,4-Bis(4-methoxyphenyl)cyclobut-3-ene-1,2-dione* (**1d**). Yellow solid, mp 173.6–176.4 °C; 0.912 g, yield 62%; ^1^H NMR (500 MHz, CDCl_3_) δ: 8.11 (d, *J* = 8.8 Hz, 4H), 7.04 (d, *J* = 8.8 Hz, 4H), 3.91 (s, 6H); ^13^C NMR (125 MHz, CDCl_3_) δ: 196.27, 184.38, 163.47, 130.40, 121.23, 114.71, 55.58; HRMS (ESI) *m*/*z* calcd for C_18_H_14_O_4_Na^+^ [M+Na]^+^: 317.07843, found: 317.07846.

*3,4-Bis(4-fluorophenyl)cyclobut-3-ene-1,2-dione* (**1e**). Yellow solid, mp 172.4–174.0 °C; 0.851 g, yield 63%; ^1^H NMR (500 MHz, CDCl_3_) δ: 8.15–8.07 (m, 4H), 7.30–7.23 (m, 4H); ^13^C NMR (125 MHz, CDCl_3_) δ: 195.56, 185.52, 165.53 (d, *J* = 256.2 Hz), 130.75 (d, *J* = 8.7 Hz), 124.40 (d, *J* = 3.8 Hz), 116.93 (d, *J* = 22.5 Hz); HRMS (ESI) *m*/*z* calcd for C_16_H_8_F_2_O_2_Na^+^ [M+Na]^+^: 293.03846, found: 293.03851.

*3,4-Bis(4-chlorophenyl)cyclobut-3-ene-1,2-dione* (**1f**). Yellow solid, mp 138.2–141.3 °C; 0.712 g, yield 47%; ^1^H NMR (500 MHz, CDCl_3_) δ: 8.00 (d, *J* = 8.0 Hz, 4H), 7.55 (d, *J* = 8.0 Hz, 4H); ^13^C NMR (125 MHz, CDCl_3_) δ: 195.32, 185.91, 139.92, 129.90, 129.42, 126.28; HRMS (ESI) *m*/*z* calcd for C_16_H_8_Cl_2_O_2_Na^+^ [M+Na]^+^: 324.97936, found: 324.97946.

*3,4-Bis(4-bromophenyl)cyclobut-3-ene-1,2-dione* (**1g**). Yellow solid, mp 163.1–165.3 °C; 1.000 g, yield 51%; ^1^H NMR (500 MHz, CDCl_3_) δ: 7.91 (d, *J* = 8.5 Hz, 4H), 7.71 (d, *J* = 8.5 Hz, 4H); ^13^C NMR (125 MHz, CDCl_3_) δ: 195.24, 186.08, 132.89, 129.44, 128.59, 126.68; HRMS (ESI) *m*/*z* calcd for C_16_H_8_Br_2_O_2_Na^+^ [M+Na]^+^: 412.87833, found: 412.87857.

*3,4-Di-m-tolylcyclobut-3-ene-1,2-dione* (**1h**). Yellow solid, mp 113.4–115.7 °C; 0.708 g, yield 54%; ^1^H NMR (500 MHz, CDCl_3_) δ: 7.92 (s, 2H), 7.85 (d, *J* = 6.8 Hz, 2H), 7.46–7.39 (m, 4H), 2.43 (s, 6H); ^13^C NMR (125 MHz, CDCl_3_) δ: 196.34, 187.65, 139.21, 134.16, 129.12, 128.76, 128.17, 125.24, 21.34; HRMS (ESI) *m*/*z* calcd for C_18_H_14_O_2_Na^+^ [M+Na]^+^: 285.08860, found: 285.08878.

*3,4-Bis(3-methoxyphenyl)cyclobut-3-ene-1,2-dione* (**1i**). Yellow solid, mp 118.0–119.5 °C; 0.706 g, yield 48%; ^1^H NMR (500 MHz, CDCl_3_) δ: 7.66 (d, *J* = 7.5 Hz, 2H), 7.63–7.59 (m, 2H), 7.45 (t, *J* = 8.0 Hz, 2H), 7.17–7.11 (m, 2H), 3.84 (s, 6H); ^13^C NMR (125 MHz, CDCl_3_) δ: 196.03, 187.45, 159.94, 130.32, 129.15, 120.69, 119.70, 112.79, 55.48; HRMS (ESI) *m*/*z* calcd for C_18_H_14_O_4_Na^+^ [M+Na]^+^: 317.07843, found: 317.07849.

*5-Dimethylamino-3,4-diphenylfuran-2(5H)-one* (**3aa**). Yellow solid, mp 145.1–146.9 °C; 0.025 g, yield 60%; ^1^H NMR (500 MHz, CDCl_3_) δ: 7.43–7.36 (m, 4H), 7.34–7.24 (m, 6H), 6.04 (s, 1H), 2.43 (s, 6H); ^13^C NMR (125 MHz, CDCl_3_) δ: 171.09, 154.09, 130.76, 129.91, 129.85, 129.40, 129.22, 128.61, 128.50, 128.34, 128.32, 97.81, 38.57; HRMS (ESI) *m*/*z* calcd for C_18_H_17_NO_2_Na^+^ [M+Na]^+^: 302.11515, found: 302.11511.

*5-Diethylamino-3,4-diphenylfuran-2(5H)-one* (**3ab**). Yellow solid, mp 117.4–118.7 °C; 0.017 g, yield 36%; ^1^H NMR (500 MHz, CDCl_3_) δ: 7.48–7.25 (m, 10H), 6.25 (s, 1H), 2.88–2.74 (m, 4H), 1.02 (t, *J* = 7.2 Hz, 6H); ^13^C NMR (125 MHz, CDCl_3_) δ: 171.49, 154.42, 131.17, 130.29, 129.86, 129.82, 129.46, 128.81, 128.74, 128.52, 128.34, 96.88, 42.35, 13.26; HRMS (ESI) *m*/*z* calcd for C_20_H_21_NO_2_Na^+^ [M+Na]^+^: 330.14645, found: 330.14633.

*5-(Pyrrolidin-1-yl)-3,4-diphenylfuran-2(5H)-one* (**3ad**). Yellow solid, mp 142.9–144.5 °C; 0.024 g, yield 52%; ^1^H NMR (500 MHz, CDCl_3_) δ: 7.44–7.26 (m, 10H), 6.29 (s, 1H), 2.96–2.82 (m, 4H), 1.81–1.73 (m, 4H); ^13^C NMR (125 MHz, CDCl_3_) δ: 171.72, 154.87, 131.16, 130.25, 129.93, 129.44, 129.11, 128.73, 128.64, 128.50, 128.44, 94.59, 46.32, 24.48; HRMS (ESI) *m*/*z* calcd for C_20_H_19_NO_2_Na^+^ [M+Na]^+^: 328.13080, found: 328.13098.

*5-(Piperidin-1-yl)-3,4-diphehylfuran-2(5H)-one* (**3ae**). White solid, mp 118.4–119.7 °C; 0.026 g, yield 55%; ^1^H NMR (500 MHz, CDCl_3_) δ: 7.48–7.38 (m, 4H), 7.36–7.26 (m, 6H), 5.97 (s, 1H), 2.87–2.74 (m, 4H), 1.60–1.41 (m, 6H); ^13^C NMR (125 MHz, CDCl_3_) δ: 171.39, 153.67, 131.06, 130.25, 129.90, 129.62, 129.41, 128.82, 128.73, 128.50, 128.34, 98.50, 48.06, 25.76, 24.05; HRMS (ESI) *m*/*z* calcd for C_21_H_21_NO_2_Na^+^ [M+Na]^+^: 342.14645, found: 342.14667.

*5-Morpholino-3,4-diphenylfuran-2(5H)-one* (**3af**). White solid, mp 166.7–169.3 °C; 0.021 g, yield 43%; ^1^H NMR (500 MHz, CDCl_3_) δ: 7.48–7.29 (m, 10H), 5.98 (s, 1H), 3.71–3.55 (m, 4H), 2.90–2.77 (m, 4H); ^13^C NMR (125 MHz, CDCl_3_) δ: 170.99, 153.07, 130.79, 130.20, 130.08, 129.96, 129.41, 128.98, 128.72, 128.60, 128.54, 96.96, 66.71, 47.19; HRMS (ESI) *m*/*z* calcd for C_20_H_19_NO_3_Na^+^ [M+Na]^+^: 344.12571, found: 344.12549.

*5-Dimethylamino-3,4-di-(4-methylphenyl)furan-2(5H)-one* (**3ba**). Yellow solid, mp 43.8–45.6 °C; 0.027 g, yield 59%; ^1^H NMR (500 MHz, CDCl_3_) δ: 7.31 (d, *J* = 8.0 Hz, 4H), 7.15 (d, *J* = 8.0 Hz, 2H), 7.11 (d, *J* = 8.0 Hz, 2H), 6.01 (s, 1H), 2.43 (s, 6H), 2.35 (s, 3H), 2.34 (s, 3H); ^13^C NMR (125 MHz, CDCl_3_) δ: 171.65, 153.62, 140.30, 138.65, 129.23, 129.22, 129.20, 128.75, 128.55, 128.18, 127.25, 97.86, 38.73, 21.40, 21.31; HRMS (ESI) *m*/*z* calcd for C_20_H_21_NO_2_Na^+^ [M+Na]^+^: 330.14645, found: 330.14655.

*5-Dimethylamino-3,4-di-(4-ethylphenyl)furan-2(5H)-one* (**3ca**). Colorless oil; 0.026 g, yield 52%; ^1^H NMR (500 MHz, CDCl_3_) δ: 7.38–7.31 (m, 4H), 7.18 (d, *J* = 8.0 Hz, 2H), 7.13 (d, *J* = 8.0 Hz, 2H), 6.01 (s, 1H), 2.69–2.61 (m, 4H), 2.44 (s, 6H), 1.26–1.21 (m, 6H); ^13^C NMR (125 MHz, CDCl_3_) δ: 171.74, 153.54, 146.45, 144.89, 129.29, 128.74, 128.64, 128.35, 128.01, 127.99, 127.51, 97.85, 38.74, 28.66, 15.24, 14.92; HRMS (ESI) *m*/*z* calcd for C_22_H_25_NO_2_Na^+^ [M+Na]^+^: 358.17775, found: 358.17795.

*5-Dimethylamino-3,4-di-(4-methoxyphenyl)furan-2(5H)-one* (**3da**). Yellow solid, mp 46.2–48.7 °C; 0.028 g, yield 55%; ^1^H NMR (500 MHz, CDCl_3_) δ: 7.42 (d, *J* = 8.5 Hz, 2H), 7.38 (d, *J* = 9.0 Hz, 2H), 6.89 (d, *J* = 8.5 Hz, 2H), 6.82 (d, *J* = 9.0 Hz, 2H), 5.97 (s, 1H), 3.82 (s, 3H), 3.81 (s, 3H), 2.44 (s, 6H); ^13^C NMR (125 MHz, CDCl_3_) δ: 171.97, 160.81, 159.81, 152.52, 130.74, 130.32, 127.21, 123.42, 122.68, 114.04, 113.93, 97.75, 55.21, 38.70; HRMS (ESI) *m*/*z* calcd for C_20_H_21_NO_4_Na^+^ [M+Na]^+^: 362.13628, found: 362.13632.

*5-Dimethylamino-3,4-di-(4-fluorophenyl)furan-2(5H)-one* (**3ea**). Yellow solid, mp 159.4–161.7 °C; 0.021 g, yield 45%; ^1^H NMR (500 MHz, CDCl_3_) δ: 7.45–7.36 (m, 4H), 7.07–6.97 (m, 4H), 5.97 (s, 1H), 2.45 (s, 6H); ^13^C NMR (125 MHz, CDCl_3_) δ: 170.90, 163.65 (d, *J* = 250.0 Hz), 163.11 (d, *J* = 248.0 Hz), 152.87, 131.43 (d, *J* = 8.7 Hz), 130.86 (d, *J* = 7.5 Hz), 128.75, 127.07 (d, *J* = 3.5 Hz), 126.06 (d, *J* = 3.7 Hz), 115.92 (d, *J* = 21.6 Hz), 115.83 (d, *J* = 21.6 Hz), 97.98, 38.74; HRMS (ESI) *m*/*z* calcd for C_18_H_15_F_2_NO_2_Na^+^ [M+Na]^+^: 338.09631, found: 338.09647.

*5-Dimethylamino-3,4-di-(4-chlorophenyl)furan-2(5H)-one* (**3fa**). Yellow solid, mp 52.3–53.9 °C; 0.021 g, yield 40%; ^1^H NMR (500 MHz, CDCl_3_) δ: 7.37–7.29 (m, 8H), 6.00 (s, 1H), 2.44 (s, 6H); ^13^C NMR (125 MHz, CDCl_3_) δ: 170.72, 153.20, 136.49, 135.24, 130.77, 130.00, 129.11, 129.05, 129.03, 128.93, 128.14, 97.98, 38.80; HRMS (ESI) *m*/*z* calcd for C_18_H_15_Cl_2_NO_2_Na^+^ [M+Na]^+^: 370.03721, found: 370.03760.

*5-Dimethylamino-3,4-di-(4-bromophenyl)furan-2(5H)-one* (**3ga**). Yellow solid, mp 54.6–56.3 °C; 0.023 g, yield 34%; ^1^H NMR (500 MHz, CDCl_3_) δ: 7.52–7.45 (m, 4H), 7.31–7.25 (m, 4H), 5.99 (s, 1H), 2.44 (s, 6H); ^13^C NMR (125 MHz, CDCl_3_) δ: 170.61, 153.28, 132.09, 131.99, 130.99, 130.16, 129.49, 129.04, 128.58, 124.93, 123.57, 97.94, 38.82; HRMS (ESI) *m*/*z* calcd for C_18_H_15_Br_2_NO_2_Na^+^ [M+Na]^+^: 457.93618, found: 457.93671.

*5-Dimethylamino-3,4-di-(3-methylphenyl)furan-2(5H)-one* (**3ha**). Yellow solid, mp 84.9–86.1 °C; 0.025 g, yield 54%; ^1^H NMR (500 MHz, CDCl_3_) δ: 7.23–7.10 (m, 8H), 6.03 (s, 1H), 2.45 (s, 6H), 2.32 (s, 3H), 2.28 (s, 3H); ^13^C NMR (125 MHz, CDCl_3_) δ: 171.51, 154.34, 138.11, 130.94, 130.80, 130.00, 129.87, 129.62, 129.53, 129.08, 128.35, 128.28, 126.42, 125.83, 97.94, 38.78, 21.37, 21.34; HRMS (ESI) *m*/*z* calcd for C_20_H_21_NO_2_Na^+^ [M+Na]^+^: 330.14645, found: 330.14642.

*5-Dimethylamino-3,4-di-(3-methoxyphenyl)furan-2(5H)-one* (**3ia**). Yellow solid, mp 113.5–115.2 °C; 0.026 g, yield 51%; ^1^H NMR (500 MHz, CDCl_3_) δ: 7.29–7.20 (m, 2H), 7.02–6.93 (m, 4H), 6.93–6.87 (m, 2H), 6.02 (s, 1H), 3.74 (s, 3H), 3.65 (s, 3H), 2.46 (s, 6H); ^13^C NMR (125 MHz, CDCl3) δ: 171.21, 159.61, 159.38, 154.16, 132.08, 131.36, 129.83, 129.63, 121.84, 121.18, 115.84, 114.91, 114.65, 114.12, 98.01, 55.24, 55.08, 38.86; HRMS (ESI) *m*/*z* calcd for C_20_H_21_NO_4_Na^+^ [M+Na]^+^: 362.13628, found: 362.13654.

*5-(Pyrrolidin-1-yl)-3,4-di-(4-methylphenyl)furan-2(5H)-one* (**3bd**). Yellow solid, mp 38.5–40.2 °C; 0.023 g, yield 45%; ^1^H NMR (500 MHz, CDCl_3_) δ: 7.33 (d, *J* = 8.0 Hz, 2H), 7.30 (d, *J* = 8.0 Hz, 2H), 7.15 (d, *J* = 7.5 Hz, 2H), 7.09 (d, *J* = 8.0 Hz, 2H), 6.24 (s, 1H), 2.96–2.80 (m, 4H), 2.35 (s, 3H), 2.34 (s, 3H), 1.80–1.73 (m, 4H); ^13^C NMR (125 MHz, CDCl_3_) δ: 172.08, 154.29, 140.18, 138.58, 129.29, 129.23, 129.15, 128.55, 128.42, 128.26, 127.51, 94.48, 46.29, 24.49, 21.43, 21.35; HRMS (ESI) *m*/*z* calcd for C_22_H_23_NO_2_Na^+^ [M+Na]^+^: 356.16210, found: 356.16229.

*5-(Piperidin-1-yl)-3,4-di-(4-methylphenyl)furan-2(5H)-one* (**3be**). White solid, mp 56.7–58.9 °C; 0.028 g, yield 54%; ^1^H NMR (500 MHz, CDCl_3_) δ: 7.38 (d, *J* = 8.5 Hz, 2H), 7.30 (d, *J* = 8.0 Hz, 2H), 7.15 (d, *J* = 8.0 Hz, 2H), 7.09 (d, *J* = 8.5 Hz, 2H), 5.93 (s, 1H), 2.87–2.71 (m, 4H), 2.35 (s, 3H), 2.34 (s, 3H), 1.58–1.42 (m, 6H); ^13^C NMR (125 MHz, CDCl_3_) δ: 171.74, 153.03, 140.13, 138.55, 129.23, 129.20, 129.02, 128.73, 128.26, 127.52, 98.38, 47.99, 25.78, 24.07, 21.39, 21.31; HRMS (ESI) *m*/*z* calcd for C_23_H_25_NO_2_Na^+^ [M+Na]^+^: 370.17775, found: 370.17786.

*5-Morpholino-3,4-di-(4-methylphenyl)furan-2(5H)-one* (**3bf**). White solid, mp 78.9–81.4 °C; 0.026 g, yield 49%; ^1^H NMR (500 MHz, CDCl_3_) δ: 7.35 (d, *J* = 8.5 Hz, 2H), 7.30 (d, *J* = 8.0 Hz, 2H), 7.16 (d, *J* = 8.5 Hz, 2H), 7.11 (d, *J* = 8.0 Hz, 2H), 5.95 (s, 1H), 3.68–3.56 (m, 4H), 2.87–2.77 (m, 4H), 2.36 (s, 3H), 2.35 (s, 3H); ^13^C NMR (125 MHz, CDCl_3_) δ: 171.37, 152.44, 140.50, 138.84, 129.28, 129.20, 129.19, 129.09, 128.61, 127.92, 127.17, 96.83, 66.71, 47.08, 21.43, 21.34; HRMS (ESI) *m*/*z* calcd for C_22_H_23_NO_3_Na^+^ [M+Na]^+^: 372.15701, found: 372.15720.

*5-(Piperidin-1-yl)-3,4-di-(4-ethylphenyl)furan-2(5H)-one* (**3ce**). Yellow oil; 0.035 g, yield 62%; ^1^H NMR (500 MHz, CDCl_3_) δ: 7.44 (d, *J* = 8.0 Hz, 2H), 7.34 (d, *J* = 8.0 Hz, 2H), 7.18 (d, *J* = 8.0 Hz, 2H), 7.11 (d, *J* = 8.5 Hz, 2H), 5.92 (s, 1H), 2.88–2.72 (m, 4H), 2.69–2.60 (m, 4H), 1.59–1.42 (m, 6H), 1.28–1.20 (m, 6H); ^13^C NMR (125 MHz, CDCl_3_) δ: 171.84, 152.92, 146.35, 144.81, 129.32, 128.84, 128.75, 128.48, 128.02, 127.78, 98.40, 48.01, 28.66, 25.81, 24.10, 15.25, 14.97; HRMS (ESI) *m*/*z* calcd for C_25_H_29_NO_2_Na^+^ [M+Na]^+^: 398.20905, found: 398.20908.

*5-Morpholino-3,4-di-(4-ethylphenyl)furan-2(5H)-one* (**3cf**). White solid; mp 51.4–53.7 °C; 0.030 g, yield 52%; ^1^H NMR (500 MHz, CDCl_3_) δ: 7.41 (d, *J* = 8.0 Hz, 2H), 7.34 (d, *J* = 8.5 Hz, 2H), 7.19 (d, *J* = 8.0 Hz, 2H), 7.14 (d, *J* = 8.5 Hz, 2H), 5.95 (s, 1H), 3.71–3.57 (m, 4H), 2.89–2.78 (m, 4H), 2.70–2.61 (m, 4H), 1.26–1.22 (m, 6H); ^13^C NMR (125 MHz, CDCl_3_) δ: 171.41, 152.32, 146.68, 145.06, 129.25, 129.07, 128.69, 128.08, 127.94, 127.39, 96.76, 66.67, 47.07, 28.66, 15.21, 14.93; HRMS (ESI) *m*/*z* calcd for C_24_H_27_NO_3_Na^+^ [M+Na]^+^: 400.18831, found: 400.18829.

*5-(Pyrrolidin-1-yl)-3,4-di-(4-methoxyphenyl)furan-2(5H)-one* (**3dd**). Yellow solid, mp 46.7–48.8 °C; 0.024 g, yield 43%; ^1^H NMR (500 MHz, CDCl_3_) δ: 7.44 (d, *J* = 9.0 Hz, 2H), 7.37 (d, *J* = 8.5 Hz, 2H), 6.89 (d, *J* = 9.0 Hz, 2H), 6.81 (d, *J* = 9.0 Hz, 2H), 6.21 (s, 1H), 3.82 (s, 3H), 3.81 (s, 3H), 2.95–2.81 (m, 4H), 1.82–1.73 (m, 4H); ^13^C NMR (125 MHz, CDCl_3_) δ: 172.35, 160.76, 159.77, 153.22, 130.79, 130.28, 126.69, 123.67, 122.92, 114.05, 113.84, 94.30, 55.22, 46.23, 24.47; HRMS (ESI) *m*/*z* calcd for C_22_H_23_NO_4_Na^+^ [M+Na]^+^: 388.15193, found: 388.15204.

*5-(Piperidin-1-yl)-3,4-di-(4-methoxyphenyl)furan-2(5H)-one* (**3de**). Yellow solid, mp 54.6–56.8 °C; 0.026 g, yield 46%; ^1^H NMR (500 MHz, CDCl_3_) δ: 7.54 (d, *J* = 9.0 Hz, 2H), 7.41 (d, *J* = 9.0 Hz, 2H), 6.93 (d, *J* = 9.0 Hz, 2H), 6.86 (d, *J* = 9.0 Hz, 2H), 5.94 (s, 1H), 3.82 (s, 3H), 3.80 (s, 3H), 2.90–2.76 (m, 4H), 1.64–1.46 (m, 6H); ^13^C NMR (125 MHz, CDCl_3_) δ: 172.01, 160.72, 159.76, 151.96, 130.76, 130.50, 127.20, 123.59, 122.95, 114.04, 113.72, 98.25, 55.20, 47.97, 25.85, 24.11; HRMS (ESI) *m*/*z* calcd for C_23_H_25_NO_4_Na^+^ [M+Na]^+^: 402.16758, found: 402.16757.

*5-Morpholino-3,4-di-(4-methoxyphenyl)furan-2(5H)-one* (**3df**). White solid, mp 65.4–67.2 °C; 0.032 g, yield 55%; ^1^H NMR (500 MHz, CDCl_3_) δ: 7.47 (d, *J* = 9.0 Hz, 2H), 7.37 (d, *J* = 9.0 Hz, 2H), 6.90 (d, *J* = 9.0 Hz, 2H), 6.83 (d, *J* = 9.0 Hz, 2H), 5.91 (s, 1H), 3.83 (s, 3H), 3.82 (s, 3H), 3.70–3.58 (m, 4H), 2.87–2.77 (m, 4H); ^13^C NMR (125 MHz, CDCl_3_) δ: 171.64, 160.92, 159.92, 151.33, 130.74, 130.38, 127.52, 123.21, 122.57, 114.11, 113.89, 96.70, 66.76, 55.25, 55.23, 47.07; HRMS (ESI) *m*/*z* calcd for C_22_H_23_NO_5_Na^+^ [M+Na]^+^: 404.14684, found: 404.14703.

*5-(Piperidin-1-yl)-3,4-di-(4-fluorophenyl)furan-2(5H)-one* (**3ee**). White solid, mp 165.6–167.9 °C; 0.019 g, yield 36%; ^1^H NMR (500 MHz, CDCl_3_) δ: 7.51–7.43 (m, 2H), 7.42–7.36 (m, 2H), 7.09–6.98 (m, 4H), 5.93 (s, 1H), 2.85–2.73 (m, 4H), 1.59–1.44 (m, 6H); ^13^C NMR (125 MHz, CDCl_3_) δ: 171.14, 163.47 (d, *J* = 250.7 Hz), 162.94 (d, *J* = 248.0 Hz), 152.46, 131.38 (d, *J* = 8.7 Hz), 130.97 (d, *J* = 8.7 Hz), 129.33 (d, *J* = 8.7 Hz), 128.48, 126.97 (d, *J* = 3.7 Hz), 126.04 (d, *J* = 3.7 Hz), 116.24 (d, *J* = 22.5 Hz), 115.82 (d, *J* = 21.2 Hz), 115.73 (d, *J* = 21.2 Hz), 98.42, 48.07, 25.77, 24.00; HRMS (ESI) *m*/*z* calcd for C_21_H_19_F_2_NO_2_Na^+^ [M+Na]^+^: 378.12761, found: 378.12781.

*5-Morpholino-3,4-di-(3-methylphenyl)furan-2(5H)-one* (**3hf**). Colorless oil; 0.022 g, yield 42%; ^1^H NMR (500 MHz, CDCl_3_) δ: 7.27–7.11 (m, 8H), 5.95 (s, 1H), 3.69–3.57 (m, 4H), 2.88–2.79 (m, 4H), 2.32 (s, 3H), 2.28 (s, 3H); ^13^C NMR (125 MHz, CDCl_3_) δ: 171.26, 153.15, 138.29, 138.12, 130.99, 130.79, 130.09, 130.00, 129.92, 129.76, 129.27, 128.45, 128.39, 126.47, 125.94, 96.99, 66.81, 47.21, 21.44, 21.42; HRMS (ESI) *m*/*z* calcd for C_22_H_23_NO_3_Na^+^ [M+Na]^+^: 372.15701, found: 372.15707.

*5-Hydroxy-3,4-diphenylfuran-2(5H)-one* (**4**). White solid, mp 146.2–147.8 °C; 0.084 g, yield 66%; ^1^H NMR (500 MHz, CD_3_OD) δ: 7.45–7.43 (m, 2H), 7.38–7.35 (m, 6H), 7.33–7.28 (m, 2H), 6.55 (s, 1H); ^13^C NMR (125 MHz, CD_3_OD) δ: 172.83, 157.84, 132.01, 131.39, 131.25, 130.45, 129.95, 129.93, 129.61, 128.97, 99.00; HRMS (ESI) *m*/*z* calcd for C_16_H_12_O_3_Na^+^ [M+Na]^+^: 275.06787, found: 275.06790.

*5-Allyl-3,4-diphenylfuran-2(5H)-one* (**5**). White solid, mp 101.3–102.7 °C; 0.028 g, yield 80%; ^1^H NMR (500 MHz, CDCl_3_) δ: 7.40–7.31 (m, 8H), 7.25–7.21 (m, 2H), 5.77–5.66 (m, 1H), 5.54–5.49 (m, 1H), 5.13–5.09 (m, 1H), 5.06–5.00 (m, 1H), 2.72–2.64 (m, 1H), 2.34–2.25 (m, 1H); ^13^C NMR (125 MHz, CDCl_3_) δ: 172.26, 159.58, 131.16, 130.63, 130.06, 129.75, 129.25, 128.99, 128.64, 128.45, 128.08, 127.25, 119.55, 80.62, 36.51; HRMS (ESI) *m*/*z* calcd for C_19_H_17_O_2_^+^ [M+H]^+^: 277.12231, found: 277.12292.

*Ethyl 2-(5-oxo-3,4-diphenyl-2,5-dihydrofuran-2-yl)acetate* (**6**). White solid, mp 77.2–78.6 °C; 0.022 g, yield 58%; ^1^H NMR (500 MHz, CDCl_3_) δ: 7.41–7.23 (m, 10H), 5.87 (dd, *J* = 9.0, 3.5 Hz, 1H), 4.19–4.10 (m, 2H), 2.76 (dd, *J* = 16.0, 3.5 Hz, 1H), 2.51 (dd, *J* = 16.0, 9.0 Hz, 1H), 1.24 (t, *J* = 7.0 Hz, 3H); ^13^C NMR (125 MHz, CDCl_3_) δ: 171.75, 169.21, 158.98, 130.62, 130.28, 129.52, 129.22, 129.14, 128.82, 128.49, 128.07, 127.13, 77.84, 61.23, 38.22, 14.09.; HRMS (ESI) *m*/*z* calcd for C_20_H_18_O_4_Na^+^ [M+Na]^+^: 345.10973, found: 345.10986.

## 4. Conclusions

We have successfully presented a silver catalyzed ring-opening [3 + 2] cycloaddition of cyclobutenediones with formamides to prepare *γ*-aminobutenolide, in which cyclobutenediones likely proceed with a key decarbonylative process. In addition, harsh reaction conditions showed the thermochemical stability of cyclobutenedione and made us pay more attention to its mechanism. Further studies including DFT calculations about the decarbonylative process and the applications to address complex synthetic issues are in progress.

## Data Availability

Crystallographic data for **3aa** (CCDC 2036646) has been deposited in the Cambridge Crystallographic Data Centre. The data can be obtained free of charge from The Cambridge Crystallographic Data Centre via www.ccdc.cam.ac.uk/data_request/cif.

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
