# Peer review of "Silver Catalyzed Decarbonylative [3 + 2] Cycloaddition of Cyclobutenediones and Formamides"

_molecules, 2021, doi:10.3390/molecules26102974_

Round 1
Reviewer 1 Report
Dear Authors,
Manuscript ID: molecules-1202617 entitled Silver Catalyzed Decarbonylative [3 + 2] Cycloaddition of Cyclobutenediones and Formamides shows synthesis of new γ-aminobutenolides. The authors done a lot of work in the field of synthesis, purification and chemical characteristics (spectral data) of the obtained compounds. A probable mechanism for obtaining the title compounds was also proposed. In my opinion, the paper should be published after minor revision (see comments).
Authors should take into account the following comments:
Please improve Introduction and Conclusion- are too laconic.
Please correct scheme (Table 2). Compound 1 into 1a-i and 2 into 2a-f.
Author Response
Responses to reviewer 1:
(1) Please improve Introduction and Conclusion- are too laconic.
˂˂Response˃˃: The sections of Introduction and Conclusion are re-written.
Added in the Introduction: As part of the transformation of small ring compounds, especially squaric acid in our group, we wish to establish an alternative approach for the construction of N,O-acetal from squaric acid or other four-membered cyclic compounds. Although the transformation of squaric acid and other cyclobutenediones to cyclopropenones under photolysis is known,[26-28] the investigation of their thermal stability is still lacking.
Added in the Conclusion: In addition, harsh reaction conditions showed the thermochemical stability of cyclobutenedione and made us pay more attention to its mechanism. Further studies including DFT calculations about the decarbonylative process and the applications to address complex synthetic issues are in progress.
(2) Please correct scheme (Table 2). Compound 1 into 1a-i and 2 into 2a-f.
Ë‚Ë‚Response˃˃: Compound number “1” is changed into “1a-i” and “2” into “2a-f” in Table 2.
Similar changes are made in Table 3, “1” into “1b-e, 1h” and “2” into “2d-f”.
Reviewer 2 Report
Given that two very similar silver-catalyzed [3+2]-cycloaddition of formamides with cyclopropenones have been developed to deliver identical products (Refs 24 & 25), the authors need to establish the advantage of their method with cyclobutenediones before this work is suitable for publication. The previous methods seem utilizing more readily available substrates (cyclopropenones v.s. cyclobutenediones), cheaper silver catalysts, lower catalyst loadings, cheaper solvents or neat, and lower reaction temperatures, also with generally higher yields and without release of carbon monoxide.
Also, no Supporting Information can be found via the online system.
Author Response
Responses to reviewer 2:
(1) Given that two very similar silver-catalyzed [3+2]-cycloaddition of formamides with cyclopropenones have been developed to deliver identical products (Refs 24 & 25), the authors need to establish the advantage of their method with cyclobutenediones before this work is suitable for publication. The previous methods seem utilizing more readily available substrates (cyclopropenones v.s. cyclobutenediones), cheaper silver catalysts, lower catalyst loadings, cheaper solvents or neat, and lower reaction temperatures, also with generally higher yields and without release of carbon monoxide.
˂˂Response˃˃: Our group is focused on the chemical transformation of four-membered cyclic compounds. In this paper, we just want to provide an alternative synthetic method to prepare 5-aminofuran-2(5H)-one which is a very important unit in natural and pharmaceutical products. We also think that the theoretical study of the reaction mechanism is necessary and interesting.
These opinions are explained in our revised manuscript (Introduction and Conclusion).
Added in the Introduction: As part of the transformation of small ring compounds, especially squaric acid in our group, we wish to establish an alternative approach for the construction of N,O-acetal from squaric acid or other four-membered cyclic compounds. Although the transformation of squaric acid and other cyclobutenediones to cyclopropenones under photolysis is known,[26-28] the investigation of their thermal stability is still lacking.
Added in the Conclusion: In addition, harsh reaction conditions showed the thermochemical stability of cyclobutenedione and made us pay more attention to its mechanism. Further studies including DFT calculations about the decarbonylative process and the applications to address complex synthetic issues are in progress.
(2) Also, no Supporting Information can be found via the online system.
˂˂Response˃˃: Supporting Information is uploaded.
Round 2
Reviewer 2 Report
N/A